# TIPE drives a cancer stem-like phenotype by promoting glycolysis via PKM2/HIF-1α axis in melanoma

**Maojin Tian[1†], Le Yang[2†], Ziqian Zhao[3†], Jigang Li[1], Lianqing Wang[1], Qingqing Yin[2], Wei Hu[1], Yunwei Lou[4], Jianxin Du[1]\*, Peiqing Zhao[1]\***

[1]Center of Translational Medicine, Zibo Central Hospital Affiliated to Binzhou Medical University, Zibo, China; [2]Shandong First Medical University, Jinan, China; [3]The Second Medical College, Xinjiang Medical University, Urumqi, China; [4]School of Laboratory Medicine, Xinxiang Medical University, Xinxiang, China

**\*For correspondence:**
jianxindu2005@126.com (JD);
bzjzzpq@163.com (PZ)

[†]These authors contributed equally to this work

## eLife Assessment

This **important** study investigates the molecular mechanisms underpinning how the tumor necrosis factor alpha-induced protein (TIPE) regulates aerobic glycolysis to promote tumor growth in melanoma. **Convincing** data using multiple independent approaches provides new insights into the molecular mechanisms underpinning aerobic glycolysis in melanoma cells. The work will be of interest to biomedical researchers working in cancer and metabolism.

**Abstract** TIPE (*TNFAIP8*) has been identified as an oncogene and participates in tumor biology. However, how its role in the metabolism of tumor cells during melanoma development remains unclear. Here, we demonstrated that TIPE promoted glycolysis by interacting with pyruvate kinase M2 (PKM2) in melanoma. We found that TIPE-induced PKM2 dimerization, thereby facilitating its translocation from the cytoplasm to the nucleus. TIPE-mediated PKM2 dimerization consequently promoted HIF-1α activation and glycolysis, which contributed to melanoma progression and increased its stemness features. Notably, TIPE specifically phosphorylated PKM2 at Ser 37 in an extracellular signal-regulated kinase (ERK)-dependent manner. Consistently, the expression of TIPE was positively correlated with the levels of PKM2 Ser37 phosphorylation and cancer stem cell (CSC) markers in melanoma tissues from clinical samples and tumor bearing mice. In summary, our findings indicate that the TIPE/PKM2/HIF-1α signaling pathway plays a pivotal role in promoting CSC properties by facilitating the glycolysis, which would provide a promising therapeutic target for melanoma intervention.

## Introduction

Melanoma is derived from the aberrant growth of melanocytes and is classified into several subgroups such as cutaneous melanoma (CM), uveal melanoma (UM), and mucosal melanoma (MM), based on its tissue locations (*Han et al., 2021*). CM is the major subgroup of melanoma. Accumulating evidence has revealed that metabolic reprogramming contributes to the tumorigenesis and progression of melanoma. Owing to its low incidence rate, targeting metabolism might be a promising therapeutic approach for the treatment of melanoma and offers a deeper understanding of its pathobiology (*Brunner and Finley, 2023*; *Faubert et al., 2020*). However, melanoma-altered metabolic reprogramming remains poorly understood.

The Warburg effect, also known as aerobic glycolysis, is a part of metabolic reprogramming. It partly contributes to increased lactate levels regardless of oxygen availability in tumor cells, providing cancer cell tumorigenicity, progression, and chemoresistance (*Guo et al., 2022*). Pyruvate kinase isoform M2 (PKM2) is the key enzyme that irreversibly converts phosphoenolpyruvate (PEP) to pyruvate in the last step of glycolysis (*Li et al., 2016*; *Liu et al., 2021*). It is shifted between catalytically inactive dimeric and active tetrameric forms, and the PKM2 dimer predominantly serves as a key glycolytic enzyme that provides advantages for tumor progression due to the Warburg effect (*Liu et al., 2017*). Post-translational modifications of PKM2, such as FGFR1-mediated Tyr105 phosphorylation and ERK1/2-dependent Ser37 phosphorylation, can promote PKM2 from a tetrameric to dimeric form, accelerating its translocation into the nucleus (*Christofk et al., 2008*; *Li et al., 2016*). In the nucleus, PKM2 acts as a protein kinase that phosphorylates STAT3 (*Chen et al., 2022b*; *Dhanesha et al., 2022*) and Histone H3 (*Niu et al., 2020*). Interestingly, nuclear PKM2 also functions as a coactivator for several transcription factors, such as HIF-1α (*Luo et al., 2011*; *Wang et al., 2014*) or Oct4 (*Yang et al., 2018*), by which regulates metabolic gene transcription and promotes tumorigenesis or sustains cancer stem cell (CSC) populations. Furthermore, HIF-1α is constitutively activated in melanoma under both normal and hypoxic conditions and is accompanied by increased glycolysis (*D'Aguanno et al., 2021*; *Malekan et al., 2021*). Therefore, the precise mechanisms underlying PKM2 activity regulation and its function as an HIF-1α coactivator might provide a molecular basis for the melanoma targeted therapy.

The tumor necrosis factor alpha-induced protein 8 (TNFAIP8, TIPE) family is composed of four members: TIPE, TIPE1, TIPE2, and TIPE3 (*Yang et al., 2022*). Accumulating evidence suggests that TIPE is strongly associated with the development of various cancers by affecting cell proliferation, apoptosis, invasion, and metastasis (*Goldsmith et al., 2017*; *Niture et al., 2018*). Therefore, TIPE is broadly considered pro-cancerous (*Padmavathi et al., 2018*). However, we previous revealed that TIPE functions as a tumor suppressor in colitis and colitis-associated colon cancer (*Lou et al., 2022*), indicating that TIPE might serve a multifarious function in malignancies. Hitherto, the role of TIPE in melanoma remains unclear. In this study, we revealed that TIPE-mediated PKM2 Ser37 phosphorylation accelerates PKM2 translocation from the cytoplasm to the nucleus, thereby elevating its binding ability to HIF-1α and hypoxia response elements (HREs), thus promoting metabolic reprogramming. TIPE-mediated metabolic reprogramming can confer tumorigenesis and stem-like phenomena to melanoma cells. This process represents an important regulator of the TIPE family for regulating aerobic glycolysis, paving the way for cancer-specific metabolism in response to low-oxygen challenge.

## Results

### TIPE increases aerobic glycolysis in melanoma cells

To characterize how TIPE performs biological functions in melanoma, we examined its expression levels in melanoma cell lines. The results showed that A375 and A875 cells had relatively higher TIPE expression than G361 and MM96L cells. Therefore, we downregulated TIPE expression in A375 cells and overexpressed TIPE in G361 cells to perform further experiments (*Figure 1—figure supplement 1*). Furthermore, we demonstrated that TIPE promotes melanoma tumorigenesis both in vitro and in vivo (*Figure 1—figure supplement 2*).

Consequently, we performed RNA sequencing analysis by overexpressing TIPE in G361 cells (*Figure 1A*) to examine the underlying mechanisms involved in the progression of melanoma. The results revealed that the overexpression of TIPE dramatically promoted expression of genes regarding glycolysis and the HIF-1α pathway (*Figure 1B, C*). Since HIF-1α pathway participates in metabolic reprogramming, especially aerobic glycolysis as well (*Baik et al., 2019*), we assumed that TIPE promotes melanoma progression might via the regulation of metabolic reprogramming. Therefore, we performed a metabolomic analysis after interfering with TIPE in A375 cells. Interestingly, the results indicated that the aerobic glycolysis pathway, including pyruvate and lactate, was significantly inhibited in melanoma cells after TIPE silencing (*Figure 1—figure supplement 3*, *Figure 1D*), confirming that TIPE facilitates glycolytic metabolism in melanoma. Consequently, ATPase activity, lactate secretion, and ATP content assays were performed to further confirm TIPE-induced glycolytic changes. The results demonstrated that TIPE decreased ATPase activity and ATP content while promoting lactate production (*Figure 1E–G*). Additionally, we measured the extracellular acidification rate (ECAR) after

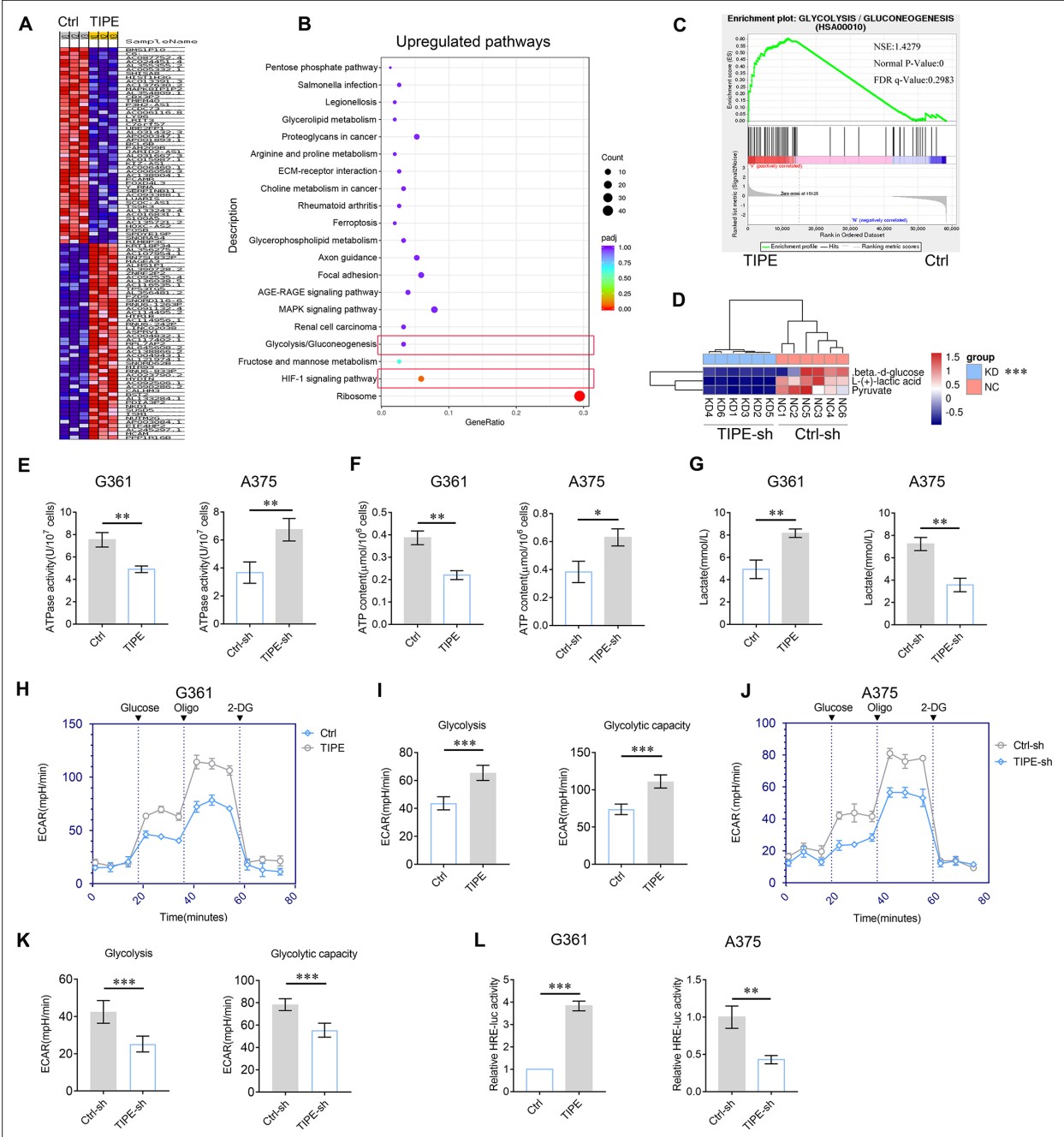

**Figure 1.** Effect of TIPE on melanoma cell glycolysis. (**A, B**) Transcriptomics analysis by unsupervised hierarchical clustering and Kyoto Encyclopedia of Genes and Genomes (KEGG) analysis showed that TIPE increased glycolysis and HIF-1α pathways. (**C**) GSEA analysis of glycolysis showed that TIPE enhanced glycolysis compared to the control. (**D**) Untargeted metabolomics analysis indicated that interfering with TIPE decreased the glycolysis pathway. (**E–G**) TIPE decreases ATPase activity and ATP content and increases lactate levels. (**H, I**) Overexpression of TIPE promotes glycolysis and glycolytic capacity according to extracellular acidification rate (ECAR) analysis. (**J, K**) Interfering with TIPE decreased glycolysis and glycolytic capacity using ECAR analysis. (**L**) TIPE significantly activated hypoxia response element (HRE) activity. *p < 0.05; **p < 0.01; ***p < 0.001. The data represent the means ± SEM of three replicates.

The online version of this article includes the following source data and figure supplement(s) for figure 1:

**Figure supplement 1.** Verification of TIPE expression in different melanoma cell lines and the effects of TIPE interference and overexpression.

**Figure supplement 1—source data 1.** Original files for western blot analysis displayed in *Figure 1—figure supplement 1*.

**Figure supplement 1—source data 2.** PDF file containing original western blots for *Figure 1—figure supplement 1*, indicating the relevant bands and treatments.

*Figure 1 continued on next page*

*Figure 1 continued*

**Figure supplement 2.** TIPE promotes melanoma cell proliferation in vitro and in vivo.

**Figure supplement 3.** Volcano plots and heatmaps generated following TIPE interference by using untargeted metabolomics.

Principal component analysis (PCA) and volcano plots of the samples from TIPE interference group vs. control in the negative mode (**a**, **b**) or positive mode (**c**, **d**) by using untargeted metabolomics. (**e**) Heatmap indicated that the glycolysis pathway including pyruvate and lactic acid is decreased after TIPE interference.

**Figure supplement 4.** TIPE enhances the expression of HIF-1α mRNA and protein.

**Figure supplement 4—source data 1.** Original files for western blot analysis displayed in *Figure 1—figure supplement 4*.

**Figure supplement 4—source data 2.** PDF file containing original western blots for *Figure 1—figure supplement 4*, indicating the relevant bands and treatments.

the overexpression or downregulation of TIPE using a Seahorse Bioscience Flux Analyzer. The results also revealed that TIPE promoted glycolysis and glycolytic capacity in melanoma cells (*Figure 1H–K*).

Furthermore, we investigated whether TIPE affected HIF-1α transcriptional activity or its expression levels. The results demonstrated that TIPE increased the expression of HIF-1α at both mRNA and protein levels (*Figure 1—figure supplement 4*). Interestingly, TIPE can dramatically activate the HRE reporter activity, with which HIF-1α is more prone to recognize the specific DNA motif to regulate metabolic gene transcription (*Chen et al., 2022a*; *Figure 1L*).

Collectively, above results demonstrate that TIPE plays a crucial role in promoting melanoma glycolysis. However, the impact of TIPE on the Warburg effect by regulating HIF-1α activity requires further investigation.

## TIPE interacts with PKM2, promotes its dimerization and nuclear import

To further study the functions of TIPE in melanoma, we used the co-immunoprecipitation mass spectrometry (Co-IP/MS) method to bait the potential binding partners of TIPE (the candidate interacting proteins of TIPE are listed in *Supplementary file 1a*). Interestingly, we found that PKM2, the key enzyme in aerobic glycolysis, might be a new partner of TIPE in melanoma (*Figure 2A* and *Figure 2— figure supplement 1*). Next, we performed Co-IP assays to confirm the interactions between TIPE and PKM2. The results revealed that exogenously overexpressed Flag-tagged PKM2 interacted with exogenously overexpressed HA-tagged TIPE (*Figure 2B, C*). Furthermore, endogenous interactions between PKM2 and TIPE were confirmed in the A375 cells (*Figure 2D, E*). GST-pull down assays demonstrated that TIPE directly bound to PKM2 in vitro (*Figure 2F*). Moreover, we performed Doulink assay to further demonstrate their endogenous interactions. The results also showed that endogenous interactions between TIPE and PKM2 were confirmed in G361 cells (*Figure 2G*). Based on these interactions, we identified the binding regions between TIPE and PKM2. We generated four fragments of PKM2, including amino acids 1–390 (NA1BA2), 45–390 (A1BA2), 1–219 (NA1B), and 220–390 (A2), based on their structural domains (*Figure 2H*). The results showed that except for the A2 domain, the NA1BA2, A1BA2, and NA1B domains of PKM2 were necessary for its interaction with TIPE (*Figure 2I*). Finally, the TIPE protein was divided into two fragments, including amino acids 1–100 (M1) and 101–198 (M2). The data showed that the amino acids 1–100 (M1) and 101–198 (M2) of TIPE interacted with PKM2 (*Figure 2J*). Due to both domains of TIPE can interact with PKM2, we believe that the A2 domain of PKM2 might have important structural effects that could affect interactions with TIPE. Taken together, these data revealed that TIPE interacts with PKM2.

Since TIPE interacts with PKM2 in melanoma, we determined whether TIPE increases aerobic glycolysis by affecting PKM2 expression or its activity. The results showed that TIPE did not influence the expression of PKM2 at either the mRNA or the protein level (*Figure 2—figure supplement 2*). Intriguingly, we demonstrated that dimeric PKM2 was reduced in TIPE-silenced A375 cells (*Figure 2K*) and was increased following TIPE overexpression in G361 cells (*Figure 2L*). Normally, a dimeric form of PKM2 with low affinity for its substrate, PEP, and a tetrameric form have a high PEP affinity. Moreover, the dimeric form of PKM2 is more prone to translocation into the nucleus and thus exerts protein kinase activity, which confers the Warburg effect and tumorigenesis (*Zhou et al., 2022*). Western blot analysis of nuclear and cytosolic fractions from A375 cells revealed that PKM2 primarily resided in the cytoplasm in which TIPE was downregulated. Likewise, the nuclear level of PKM2 was increased

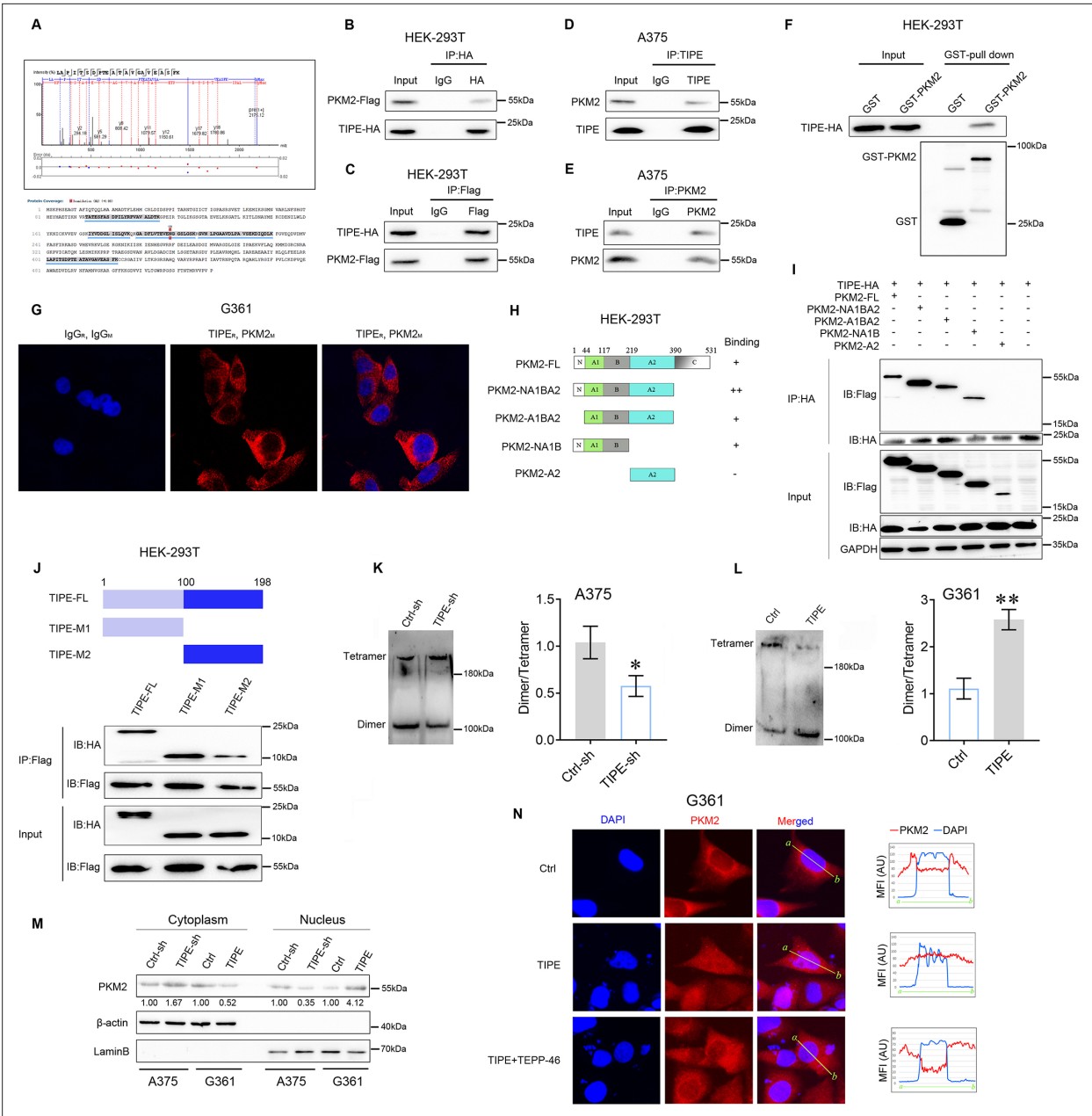

**Figure 2.** TIPE interacts with PKM2 to govern its nuclear import in a dimeric form-dependent manner. (**A**) Co-immunoprecipitation mass spectrometry (Co-IP/MS) analysis demonstrated that PKM2 interacted with TIPE in A375 cells. The results indicated that there were five peptides of PKM2 were detected to interact with TIPE (below), and one of which was shown at the upper. (**B, C**) Immunoprecipitation (IP) and western blot analysis of the exogenous TIPE/PKM2 proteins interaction in the HEK-293T cells co-transfected with Flag-tagged PKM2 and HA-tagged TIPE. (**D, E**) IP and western blot analysis of the endogenous TIPE/PKM2 proteins interaction in the A375 cells. (**F**) GST-pull down assay analysis of TIPE/PKM2 proteins interaction using purified GST-tagged PKM2 and TIPE-HA. (**G**) TIPE (rabbit source, TIPE_R) endogenously interacted with PKM2 (mouse source, PKM2_M) in G361 cells by using the Doulink assay. The results suggested that TIPE interacted with PKM2 to form a red color complex. The red color signal is generated only when the two proteins are too close to interaction. (**H, I**) IP and western blot analysis of HA-tagged TIPE and Flag-tagged PKM2 fragment protein interaction in HEK-293T cells. (**J**) IP and western blot analysis of the Flag-tagged PKM2 and HA-tagged TIPE fragment protein interaction in HEK-293T cells. (**K**) Interfering TIPE decreased PKM2 dimeric formation but increased tetramer formation, as analyzed by BN-PAGE with β-actin as a loading control. (**L**) Overexpression of TIPE increased PKM2 dimeric formation and decreased tetramer formation. (**M**) Western blot showed that TIPE promoted PKM2 translocation into the nucleus. (**N**) TIPE enhanced PKM2 translocation into the nucleus, and this phenomenon was diminished by the administration of TEPP-46 (100 μM). *p < 0.05; **p < 0.01. The data represent the means ± SEM of three replicates.

The online version of this article includes the following source data and figure supplement(s) for figure 2:

**Source data 1.** Original files for western blot analysis displayed in *Figure 2*.

*Figure 2 continued*

**Source data 2.** PDF file containing original western blots for *Figure 2*, indicating the relevant bands and treatments.

**Figure supplement 1.** TIPE engages in an interaction with PKM2.

**Figure supplement 1—source data 1.** Original files for western blot analysis displayed in *Figure 2—figure supplement 1*.

**Figure supplement 1—source data 2.** PDF file containing original western blots for *Figure 2—figure supplement 1*, indicating the relevant bands and treatments.

**Figure supplement 2.** TIPE has no effect on the expression levels of PKM2.

**Figure supplement 2—source data 1.** Original files for western blot analysis displayed in *Figure 2—figure supplement 2*.

**Figure supplement 2—source data 2.** PDF file containing original western blots for *Figure 2—figure supplement 2*, indicating the relevant bands and treatments.

after overexpression of TIPE in G361 cells (*Figure 2M*). Furthermore, we examined the distribution of PKM2 after TIPE overexpression by using a Thermo Fisher EVOS microscopic analysis. As shown in *Figure 2N*, PKM2 was predominantly present in the nucleus after TIPE overexpression, and this phenomenon was diminished by the administration of a PKM2 dimer formation suppressor (TEPP-46, 100 μM). It has been shown that dimeric PKM2 promotes tumor progression by regulating the Warburg effect (*Zhou et al., 2022*). Thus, we speculated that TIPE restores the nuclear levels of PKM2 to promote melanoma tumorigenicity via a dimeric PKM2-dependent Warburg effect.

## TIPE requires PKM2 for transcriptional activation of HIF-1α

Accumulating evidence has revealed that tetrameric PKM2 is enzymatically active toward pyruvate and acts as a transcriptional coactivator, while it dissociates into a dimer (*Wei et al., 2020*). Consequently, the dimeric form of PKM2 becomes a coactivator of HIF-1α to transcriptionally activate glycolytic genes in favor of the Warburg effect (*Figure 3A*; *Ouyang et al., 2018*). As PKM2 and HIF-1α both play important roles in the metabolic reprogramming process, it is reasonable to believe that the interactions between TIPE and these two molecules could affect the glycolysis in melanoma. Thus, we investigated whether TIPE promoted the transcriptional activation of HIF-1α in a PKM2-dependent manner.

To confirm the effect of TIPE activation on HIF-1α transcriptional activity, we tested the ability of TIPE to activate the HRE reporter activity in HRE-luciferase plasmid transfected HEK-293T cells. Not surprisingly, significant augmentation of HRE promoter activation by TIPE was observed, especially when combined with PKM2 (*Figure 3B*). Consequently, TIPE-activated HRE reporter activity in a dose-dependent manner, and this phenomenon was reversed after PKM2 knockdown, demonstrating that TIPE has critical effects on the activation of HIF-1α transcription in a PKM2-dependent manner (*Figure 3C*). HIF-1α targeted genes such as *LDHA* and *SLC2A1* were also upregulated by TIPE in a dose-dependent manner and were reversed by interfering with PKM2 (*Figure 3D, E*). Due to PKM2 cooperated with HIF-1α to activate glycolytic genes during the Warburg effect, we used the Duolink method to confirm whether TIPE affects the endogenous interaction between HIF-1α and PKM2. The results showed that the interaction between HIF-1α and PKM2 was decreased by TIPE interference, whereas the interaction was increased when TIPE was overexpressed (*Figure 3F, G*). More interestingly, TIPE enhanced the PKM2/HIF-1α interaction in the nucleus, further indicating that TIPE-induced PKM2 nuclear translocation and thus promoted its interaction with HIF-1α (*Figure 3H*). In addition, the Co-IP assay revealed that TIPE elevated this interaction in a dose-dependent manner (*Figure 3I*).

Moreover, we analyzed the correlation between TIPE and hypoxia scores using TCGA dataset. Interestingly, we found that the hypoxia score was significantly related to the expression of TIPE (*Figure 3J*; *Supplementary file 1b*), and the TIPE higher expression group had a higher hypoxia score than the lower expression group in melanoma (*Figure 3K*).

Collectively, these results demonstrated that TIPE has critical effects on the transcriptional activation of HIF-1α in a PKM2-dependent manner.

## TIPE promotes HIF-1α transcriptional activation via PKM2 Ser37 phosphorylation

The presence of PKM2 in the nucleus enhances glucose metabolism by regulating HIF-1α (*Luo et al., 2011*). Our results preliminarily demonstrate that TIPE promotes PKM2 dimerization and

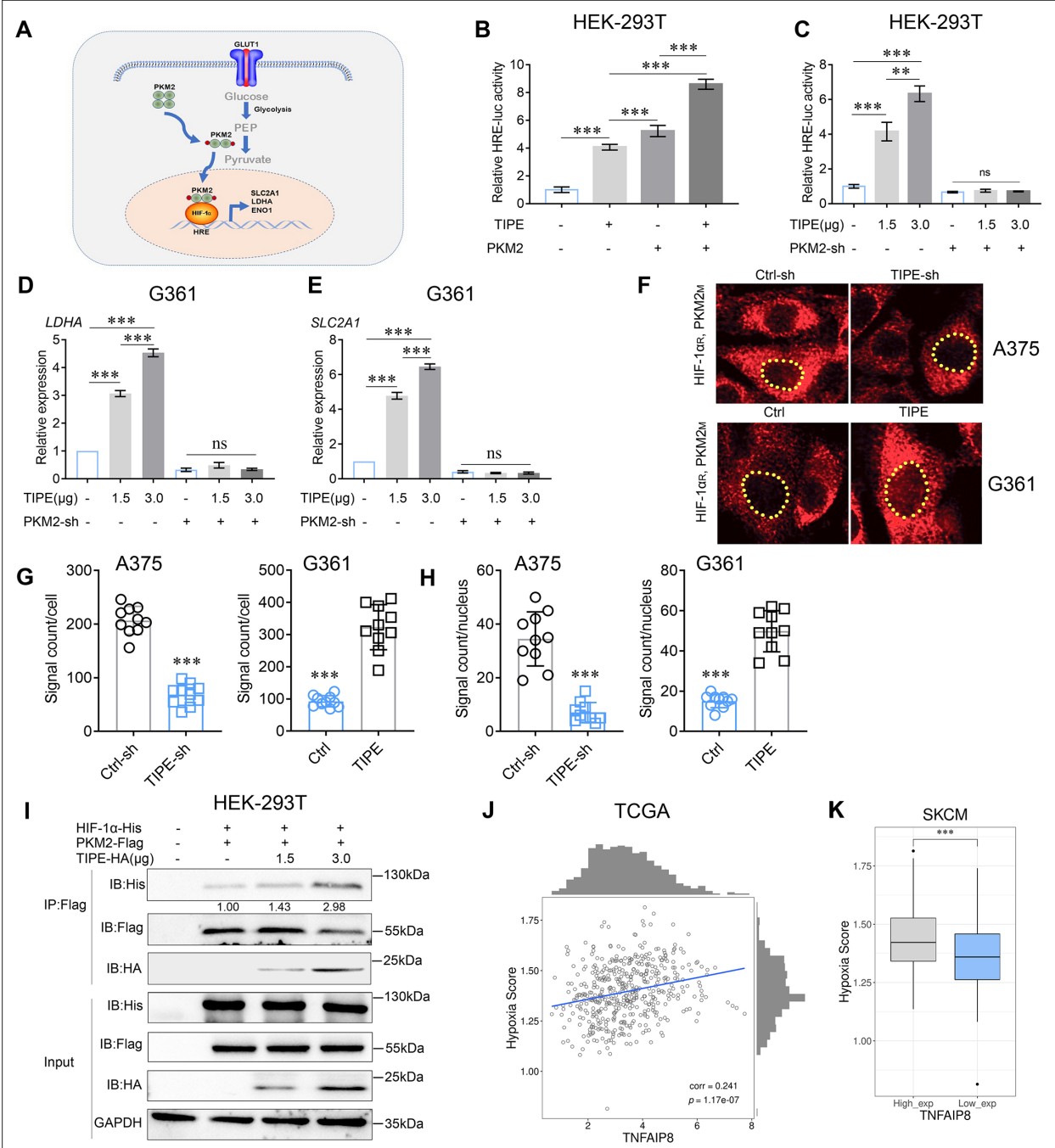

**Figure 3.** TIPE promotes HIF-1α transcription in a PKM2-dependent manner. (**A**) Proposed molecular mechanism by which dimeric PKM2 regulates cell proliferation and glycolysis by modulating HIF-1α activity. (**B**) TIPE, especially when combined with PKM2, boosts relative hypoxia response element (HRE) luciferase activity, as examined by luciferase reporter assay. (**C**) TIPE promoted HRE activity in a dose- and PKM2-dependent manner. (**D, E**) TIPE increases HIF-1α targeted genes, including *LDHA* and *SLC2A1*, in a dose- and PKM2-dependent manner. (**F, G**) TIPE promoted endogenous interaction between PKM2 and HIF-1α in melanoma cells using a Doulink assay. Interference of TIPE in A375 cells promoted the interaction between PKM2 and HIF-1α (upper) compared to that overexpression of TIPE in G361 cells decreased their interaction (lower). The density of the red color signaling means the interactive strength between PKM2 and HIF-1α affected by TIPE. (**H**) TIPE enhanced the PKM2/HIF1a interaction in the nucleus. (**I**) TIPE increased the exogenous interaction between PKM2 and HIF-1α in a dose- dependent manner in HEK-293T cells. (**J**) TCGA dataset revealed that TIPE has a positive relationship with hypoxia score in melanoma. (**K**) Higher expression of TIPE is associated with a relatively higher hypoxia score in melanoma. *p < 0.05; **p < 0.01; ***p < 0.001. The data represent the means ± SEM of three replicates *p<0.05.

The online version of this article includes the following source data for figure 3:

**Source data 1.** Original files for western blot analysis displayed in *Figure 3*.

*Figure 3 continued on next page*

*Figure 3 continued*

**Source data 2.** PDF file containing original western blots for *Figure 3*, indicating the relevant bands and treatments.

nuclear import, thereby increasing HIF-1α transcriptional activation and promoting aerobic glycolysis. Normally, Erk1/2-dependent phosphorylation of PKM2 at serine 37 (Ser37) and FGFR1-dependent phosphorylation of PKM2 at tyrosine 105 (Tyr105) (*Li et al., 2016*; *Zhou et al., 2018*) are indicative of its dimerization and translocation, further promoting the Warburg effect and serving as a transcription factor coactivator (*Novoyatleva et al., 2019*; *Wei et al., 2020*). As shown in *Figure 4A*, interference with TIPE in A375 cells blocked PKM2 Ser37 phosphorylation, and overexpression of

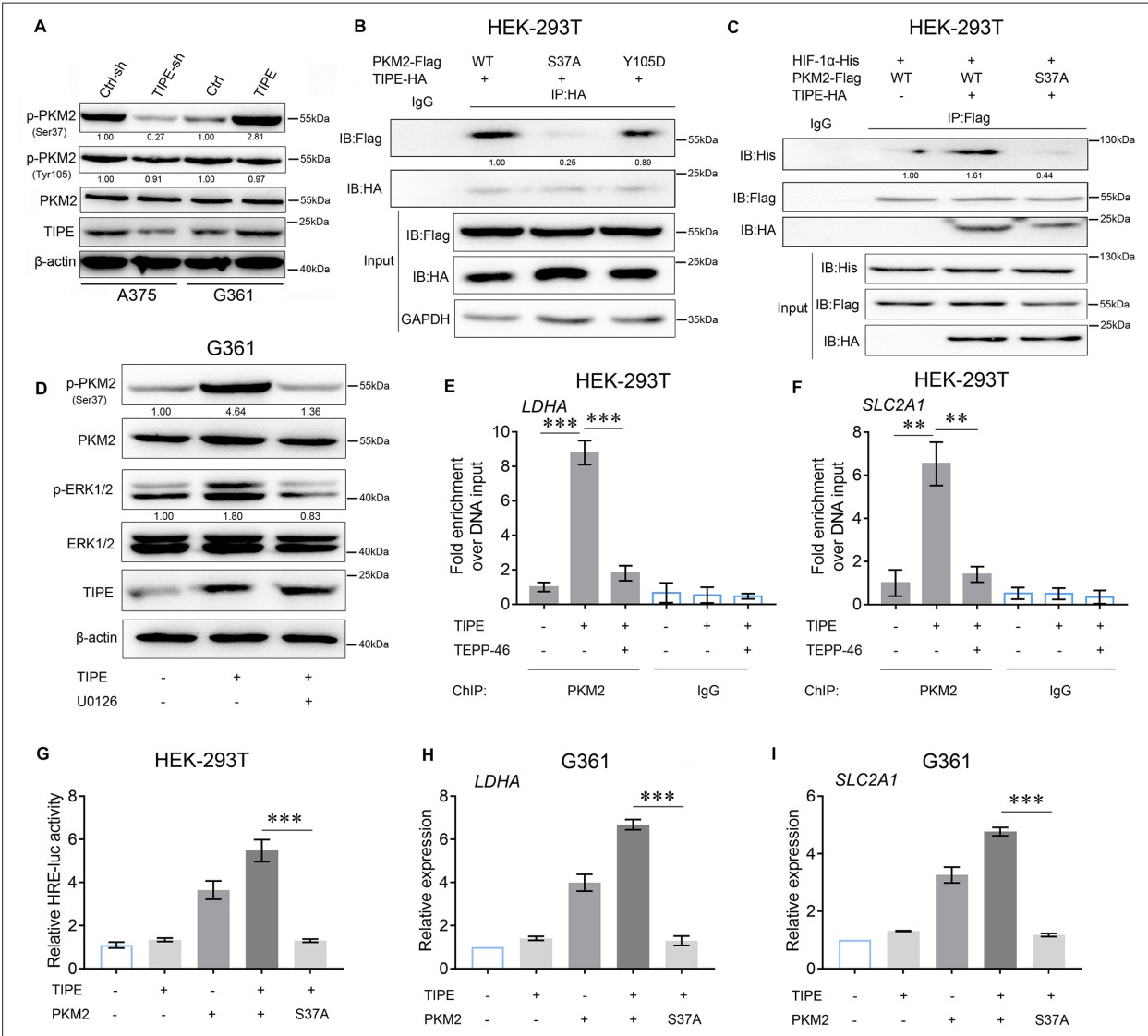

**Figure 4.** TIPE increases HIF-1α activity depending on PKM2 Ser37 phosphorylation. (**A**) TIPE enhanced PKM2 Ser37 phosphorylation, but not that of Tyr105. (**B**) PKM2 Ser37, but not Tyr105, increased its interaction with TIPE. (**C**) PKM2 Ser37 mutation (S37A) hampered its interaction with HIF-1α promoted by TIPE. (**D**) TIPE elevated PKM2 Ser37 phosphorylation in an ERK-dependent manner. (**E, F**) TIPE enhanced PKM2 binding to the hypoxia response element (HRE) for *LDHA* and *SLC2A1* in a dimeric form-dependent manner. (**G**) PKM2 Ser37 mutation (S37A) inhibited the HRE activity induced by TIPE. (**H, I**) PKM2 Ser37 mutation (S37A) decreased the expression of *LDHA* and *SLC2A1* that promoted by TIPE. **p < 0.01; ***p < 0.001. The data represent the means ± SEM of three replicates.

The online version of this article includes the following source data for figure 4:

**Source data 1.** Original files for western blot analysis displayed in *Figure 4*.

**Source data 2.** PDF file containing original western blots for *Figure 4*, indicating the relevant bands and treatments.

TIPE in G361 cells caused Ser37 phosphorylation compared to controls. However, the phosphorylation of PKM2 Tyr105 did not show any obvious changes. Consequently, we hypothesized that this transformation caused by Ser37 phosphorylation affects the interaction between TIPE and PKM2. As expected, the mutation of PKM2 at S37A blocked its interaction with TIPE, but not with the Y105D mutation (*Figure 4B*). Interestingly, this mutation also interfered with the interaction between PKM2 and HIF-1α, which was elevated by TIPE overexpression (*Figure 4C*). A previous study has revealed that TIPE promotes chemoresistance in acute myeloid leukemia (AML) by activating the ERK signaling pathway (*Pang et al., 2020*). In addition, phosphorylation of PKM2 at serine 37 is ERK1/2-dependent manner (*Yang et al., 2012*). TIPE does not serve as a phosphorylase to perform such post-translational modifications directly. Taken together, these results indicate that TIPE promotes PKM2 dimerization depending on ERK signaling. As expected, the ERK inhibitor, U0126, inhibited PKM2 Ser37 phosphorylation induced by TIPE overexpression (*Figure 4D*). To investigate whether TIPE affects the binding affinity of PKM2 to the Hif-1α-specific binding site of the *LDHA* and *SLC2A1* promoter using chromatin immunoprecipitation *q*PCR (ChIP-*q*PCR) analysis, we found that TIPE increased the binding of PKM2 to the *LDHA* and *SLC2A1* promoters. This phenomenon, caused by the overexpression of TIPE, was diminished after the administration of TEPP-46, indicating that TIPE promoted the binding of PKM2 to the *LDHA* and *SLC2A1* promoters in a dimeric PKM2-dependent manner (*Figure 4E, F*). Finally, we demonstrated that the augmentation of TIPE on HRE promoter activation or HIF-1α targeted genes such as *LDHA* and *SLC2A1* was blocked by the S37A mutation (*Figure 4G–I*).

In brief, we revealed that TIPE promoted HIF-1α transcriptional activation via ERK-dependent PKM2 Ser37 phosphorylation.

## TIPE facilitates melanoma tumorigenesis and aerobic glycolysis in a dimeric PKM2-dependent manner

Several studies have demonstrated that dimeric PKM2 increases tumor progression by regulating the Warburg effect. Therefore, we speculated that TIPE promotes melanoma growth through a dimeric PKM2-dependent Warburg effect. To demonstrate this, we knocked down TIPE in A375 cells and found that cell proliferation was decreased. Interestingly, we found that Pyridoxine, a PKM2 dimer formation inducer (*Wei et al., 2020*), was sufficient to abrogate the TIPE silencing-mediated cell proliferation (*Figure 5A*). Consistently, TEPP-46, a PKM2 dimer formation suppressor, was sufficient to reverse the increased proliferative capacity induced by TIPE overexpression in G361 cells (*Figure 5B*). Clone formation ability (*Figure 5C, D*) and in vivo experiments (*Figure 5E–G*) also demonstrated this phenomenon. Interestingly, we found that decreased ECAR, glycolysis, and glycolytic capacity induced by TIPE interference could be fully antagonized by Pyridoxine in A375 cells (*Figure 5H–J*). In addition, TEPP-46 reversed the ECAR, glycolysis, and glycolytic capacity induced by TIPE overexpression in G361 cells (*Figure 5K–M*).

Lastly, to further demonstrate how TIPE affects the PKM2/HIF-1α pathway during the Warburg effect, we revealed that inhibition of PKM2 dimerization can rescue TIPE-induced cell proliferation. However, this phenomenon was diminished by the activation of HIF-1α, demonstrating that TIPE promotes melanoma cell proliferation via dimeric PKM2-dependent HIF-1α activation (*Figure 5N, O*). More interestingly, we evidenced that suppression of PKM2 dimerization inhibited the TIPE-mediated Warburg effect. In addition, this phenomenon was diminished by the activation of HIF-1α (*Figure 5P, Q*).

Taken together, these data suggest that TIPE regulates cell growth and the Warburg effect by inducing the formation of dimeric PKM2, and thus activating the HIF-1α pathway.

## TIPE accelerates melanoma cells stem-like phenomenon via promoting PKM2 dimerization under normoxic conditions

Increasing evidence has shown that the HIF-1α pathway is associated with acquisition of cancer stem-like properties (*Weinstein et al., 2022*; *Zhu et al., 2022*). To confirm the effects of TIPE on the stemness of melanoma cells under normoxic conditions, we examined the stemness-associated features in vitro. First, we measured cancer stem-like phenotype markers, including *NANOG*, *NOTCH*, *POU5F1(OCT3/4)*, *SOX2*, and *BMI-1*. The results showed that the markers were increased after overexpression of TIPE in G361 cells and were reduced after TIPE interference in A375 cells by using *q*PCR method. More interestingly, neural crest stem cell markers (including *NES* and *SOX10*) that are

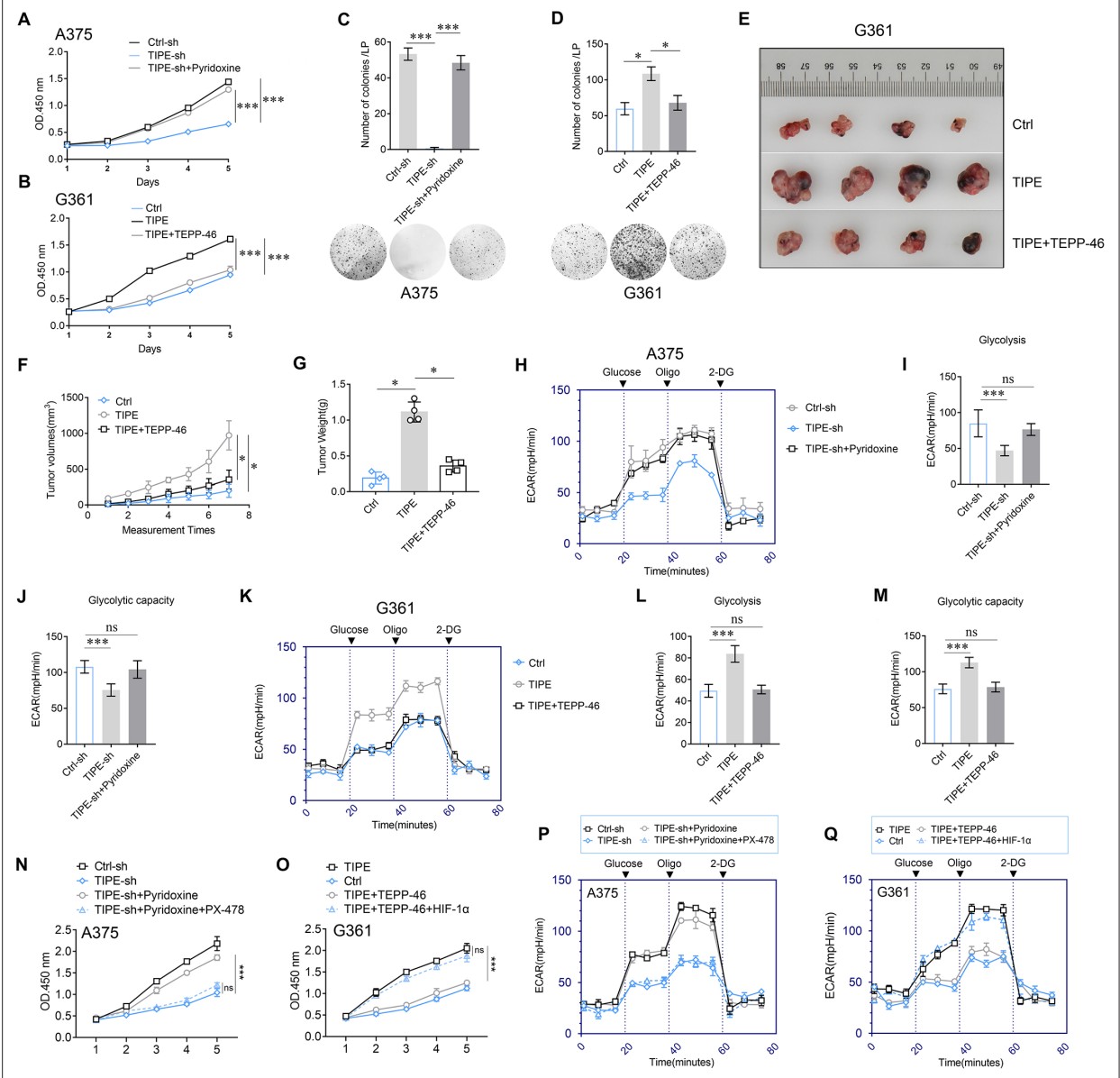

**Figure 5.** TIPE promotes melanoma cells proliferation and glycolysis depending on PKM2 dimerization. (**A**) Pyridoxine (which facilitates PKM2 dimerization) treatment reversed the inhibition of cell proliferation that induced via interfering TIPE. (**B**) Administration of TEPP-46 rescued TIPE overexpression induced cell proliferation. (**C, D**) TIPE promoted melanoma clone formation in a dimeric PKM2-dependent manner. (**E–G**) In vivo experiments showed that TIPE promoted melanoma proliferation via the dimerization of PKM2. (**H–J**) Activation of PKM2 dimerization reversed the inhibition of glycolysis and glycolytic capacity after TIPE interfering. In contrast, inhibition of PKM2 dimerization altered this phenomenon (**K–M**). (**N**) Pyridoxine increased the cell proliferation that was inhibited by TIPE-sh. It was rescued by inhibition of HIF-1α. (**O**) TEPP-46 inhibited the TIPE-induced cell proliferation, and was rescued by overexpression of HIF-1α. (**P**) Pyridoxine enhanced the Warburg effect that was suppressed by TIPE-sh, and it was diminished via administration of PX-478 (a HIF-1α inhibitor). (**Q**) TEPP-46 can suppression of the Warberg effect that was increased by overexpression of TIPE, this phenomenon was diminished by overexpression of HIF-1α. *p < 0.05; ***p < 0.001. The data represent the means ± SEM of three replicates.

more relevant to melanoma biology were also greatly changed (***Figure 6A, B***). We further revealed that TIPE increased the migration ability and increased sphere formation (***Figure 6C, D***). In addition, TIPE significantly increased the chemosensitivity of melanoma cell to sorafenib (***Figure 6E, F***). Next, we examined the tumorigenicity of melanoma cells with TIPE knockdown in vivo by injecting Ctrl-sh or TIPE-sh A375 cells subcutaneously into nude mice at three dilutions (1 × 10⁶, 1 × 10⁵, and 1 × 10⁴) and allowing them to grow for 4 weeks. The confidence intervals (CIs) for 1/(stem cell frequency)

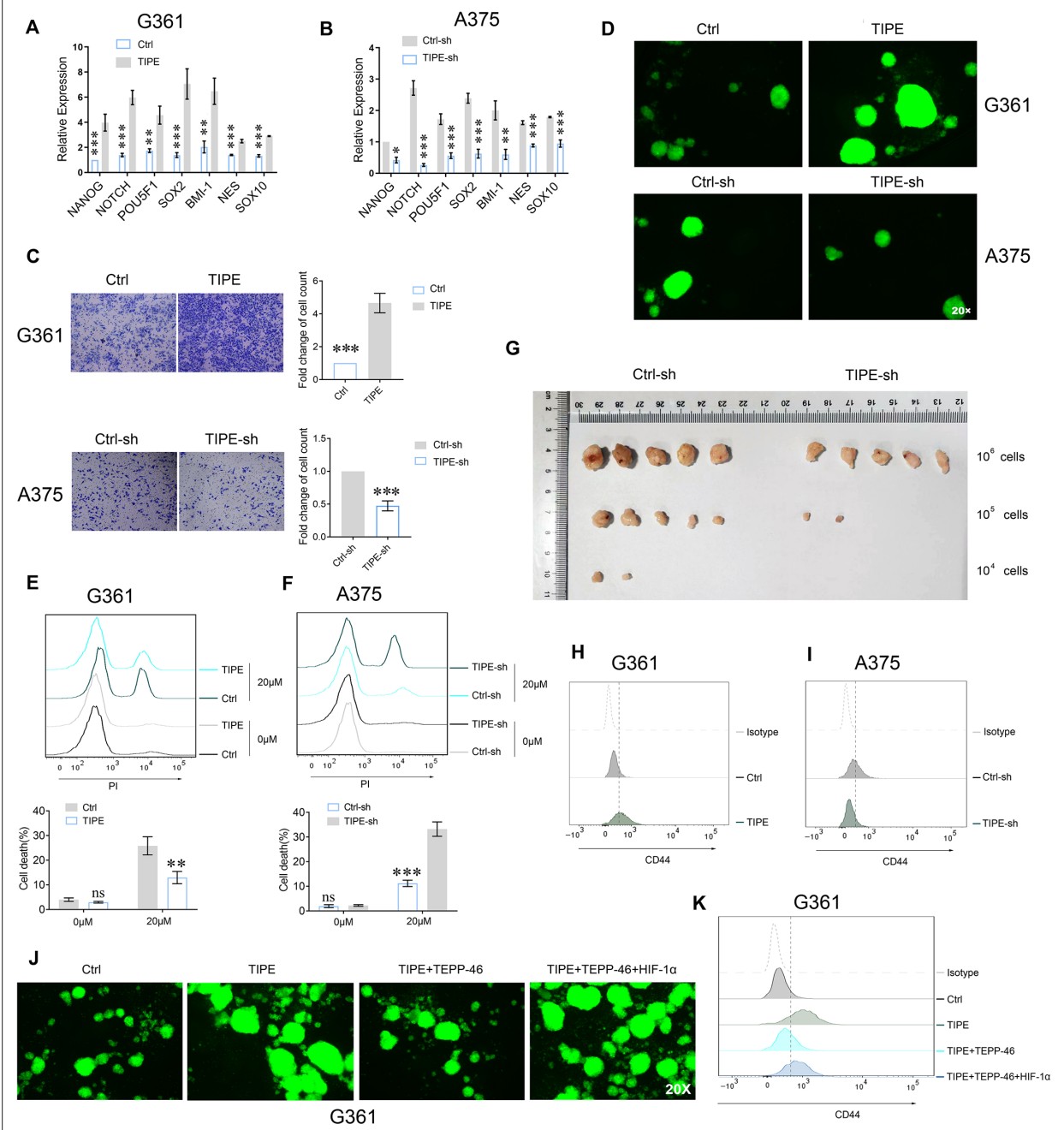

**Figure 6.** TIPE fostered PKM2 dimerization increases melanoma stem-like phenomenon. (**A, B**) TIPE increased the cancer stem-like phenotype markers, including *NANOG*, *NOTCH*, *POU5F1*, *SOX2*, *BMI-1*, *NES*, and *SOX10,* measured by *q*PCR. The effects of TIPE on the stemness of melanoma cells are shown by in vitro cell migration (**C**), sphere formation (**D**), and chemoresistance (**E, F**). (**G**) Limiting dilution xenograft formation of A375 cells with TIPE interference. Nude mice were subcutaneously injection of indicated cells. After about 60 days later, the mice were sacrificed and the confidence intervals (CIs) for 1/(stem cell frequency) were calculated. (**H, I**) TIPE increased the expression of CD44+ cells in melanoma. (**J, K**) The effects of TIPE on the stemness of melanoma cells following the administration of TEPP-46, and this phenomenon was reversed by overexpression of HIF-1α, as evidenced by sphere formation (**J**) and CD44+ cell population (**K**). *p < 0.05; **p < 0.01; ***p < 0.001. The data represent the means ± SEM of three replicates.

using extreme limiting dilution were calculated as previous reported (*Hu and Smyth, 2009*; *Niu et al., 2021*). The estimated CI for the frequency of CSCs in the TIPE knockdown group was 202,749, compared with 18,441 in the control group (p < 0.001) (*Figure 6G*; *Supplementary file 1c, d*). Moreover, CD44+ subpopulations were significantly elevated, as detected using a fluorescence-activated cell sorting (FACS) assay (*Figure 6H, I*).

Because TIPE promotes HIF-1α transcriptional activation in a dimeric PKM2-dependent manner, we performed a sphere formation assay by overexpressing TIPE in G361 cells. The results showed that TEPP-46 partially reversed tumorsphere formation induced by TIPE overexpression. Moreover, overexpression of HIF-1α diminished this phenomenon, indicating that TIPE-induced sphere formation depended on dimeric PKM2-dependent HIF-1α activation (*Figure 6J*). In addition, the increased CD44⁺ subpopulations of G361 cells induced by TIPE were decreased by the administration of TEPP-46, and could be rescued by overexpression of HIF-1α (*Figure 6K*).

These findings strongly suggest that TIPE maintains melanoma cell stemness via dimeric PKM2-dependent HIF-1α activation.

## TIPE is positively correlated with the levels of PKM2 Ser37 phosphorylation and CSC markers

We analyzed the expression of TIPE in melanoma and benign skin tumors, and the results showed that the expression of TIPE was dramatically increased compared to controls using a tissue chip (*Figure 7A, B*). Furthermore, the expression of TIPE in primary group (non-lymph node metastasis) was lower than that of the metastatic group (lymph node metastasis) (*Figure 7C*). To determine if there was a correlation between TIPE and PKM2 Ser37 levels and other indicators such as Ki67- and hypoxia-related gene (LDH), their expression were analyzed by immunohistochemistry (IHC) staining. The results showed that TIPE expression was positively correlated with PKM2 Ser37, Ki67, and LDH expression (*Figure 7D*). However, an analysis of the correlation between TIPE levels and the outcomes of melanoma patients was not conducted owing to the absence of survival data within the tissue chip. Intriguingly, the findings from the TCGA dataset present a paradoxical observation, suggesting that elevated TIPE expression is associated with a more favorable prognosis in melanoma patients (*Figure 7—figure supplement 1*). This unexpected correlation underscores the need for a more comprehensive investigation into the underlying mechanisms.

Consistent with the above findings, IHC results showed that TIPE, PKM2 Ser37, LDH, and CSCs marker CD44 were inhibited in A375 TIPE-interfering cell-derived xenografts (*Figure 7E*). Lastly, TCGA dataset revealed that TIPE has a positive relationship with CSCs markers including *BMI1*, *NANOG*, *NOTCH1*, and *POU5F1* in melanoma (*Figure 7F–I*).

These results further demonstrate that TIPE maintains CSC phenotypes in a PKM2 dimerization manner.

## Discussion

Accumulating evidence has revealed that TIPE serves as an oncogene and a negative regulator of apoptosis. Its overexpression is positively associated with the development of various cancers, including liver, lung, and breast cancers (*Lou et al., 2022*; *Niture et al., 2018*). However, Chen et al. reported that TIPE initially inhibits cancer cell proliferation but promotes tumor migration and metastasis (*Li et al., 2021*). In addition, a previous report has demonstrated that TIPE functions as a tumor suppressor in certain inflammatory carcinogenesis (*Lou et al., 2022*). These results suggest that TIPE plays a variety of roles in cancer progression. Interestingly, *Niture et al., 2021* reported that TIPE promotes prostate cancer progression by regulating oxidative phosphorylation and glycolysis . However, the mechanism through which TIPE regulates metabolic reprogramming remains unclear. Here, we demonstrate that TIPE facilitates the Warberg effect by promoting dimeric PKM2-mediated HIF-1α transactivation in melanoma. Therefore, targeting TIPE may offer a novel therapeutic approach for melanoma treatment.

Mammalian pyruvate kinase consists of four isoforms, M1, M2, L, and R, which are encoded by two distinct genes (PKM and PKLR) (*Valentini et al., 2002*). Although these isoforms share similar features, only PKM2 exhibits the lowest basal enzymatic activity and is capable of configuration transformation between the tetramer and dimer, meeting differential metabolic demands (*Morgan et al., 2013*; *Puckett et al., 2021*). Cancer cells shift toward the preferential expression of the specific isoform of PKM2, especially its dimeric form, and serve as a pivotal metabolic enzyme that promotes the activation of oncogenic pathways in tumors (*Icard et al., 2022*). Normally, PKM2 enzymatic activity is accurately modulated in a variety of ways, such as post-translational modifications or metabolite binding, to maintain balanced levels in tumor cells (*Puckett et al., 2021*). Here, we showed that TIPE promoted

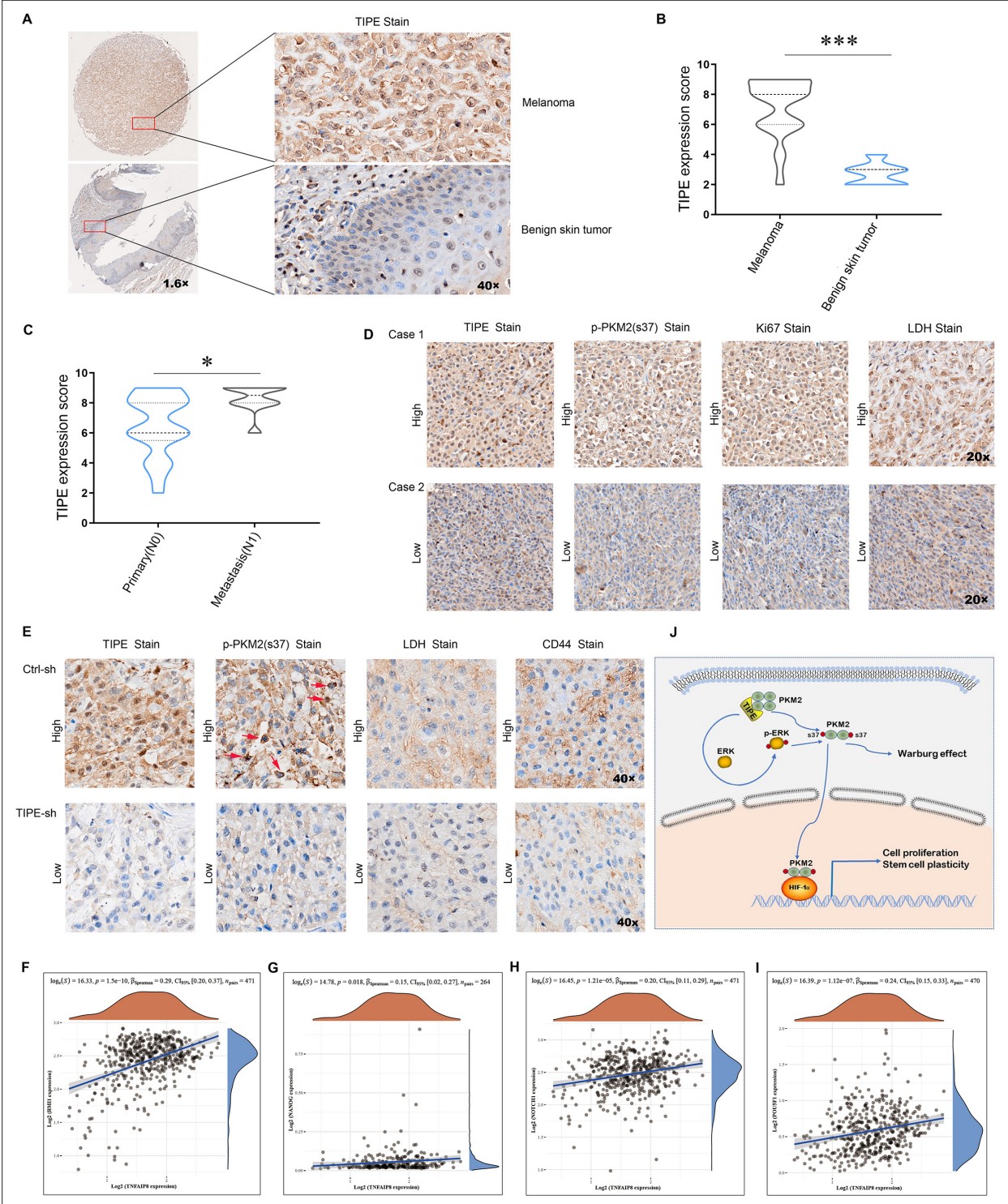

**Figure 7.** TIPE positively corelated with cancer stem cell (CSC) markers and the levels of p-PKM2(Ser37). (**A, B**) Higher expression of TIPE was observed in melanoma tumor tissues than in the control, as evidenced by immunohistochemistry. (**C, D**) The expression of TIPE correlated well with p-PKM2(Ser37) in melanoma tumor tissues. (**E**) Similarly, a good correlation was observed between TIPE, p-PKM2(Ser37), LDH, and CD44 in mouse xenografts. (**F–I**) In addition, the expression of TIPE was positively correlated with CSCs markers, including *BMI1*, *NANOG*, *NOTCH1*, and *POU5F1* in TCGA dataset. (**J**) A brief model depicting the functional impact of TIPE on metabolic reprogramming in melanoma. ***p < 0.001. The data represent the means ± SEM of three replicates "*p<0.05.

The online version of this article includes the following figure supplement(s) for figure 7:

*Figure 7 continued on next page*

*Figure 7 continued*

**Figure supplement 1.** TCGA dataset showed a paradoxical observation that elevated TIPE expression is associated with a favorable prognosis in melanoma patients.

melanoma tumorigenesis by promoting glycolysis. The results showed that TIPE interacts with PKM2 and may participate in glycolysis. However, TIPE did not influence the expression of PKM2 at either the mRNA or the protein level. Because of the modulation of PKM2 kinetic properties primarily via its dimer and tetramer configurations, we demonstrated that TIPE could promote its dimeric form transformation, increasing PKM2 residing in the nucleus. However, the downstream signaling pathways that are mediated by TIPE in melanoma remain largely unknown.

The nuclear PKM2 routinely serves as a coactivator to transcriptionally activate certain metabolic genes (*Cluntun et al., 2017*; *Zhang et al., 2019*). Because TIPE binds PKM2 and transports it into the nucleus, the dimeric form of PKM2 more easily enters the nucleus to become a coactivator of HIF-1α, resulting in the upregulation of HIF-1α targeted genes such as *LDHA* and *SLC2A1*. Thus, we elucidate these relationships in the next step. To verify whether TIPE activates HIF-1α transcriptional activity, we found that TIPE significantly increased the activation of the HRE promoter, especially when combined with PKM2. However, this activation was diminished by interfering with PKM2, indicating that TIPE has critical effects on the activation of HIF-1α transcription in a PKM2-dependent manner. Because PKM2 interacts with HIF-1α and potentiates PKM2-mediated HIF-1α transcription, we found that TIPE increased the interaction between HIF-1α and PKM2. In TCGA dataset, the hypoxia score was positively correlated with the mRNA expression of TIPE in melanoma, and the TIPE higher expression group had a higher hypoxia score than the lower expression group.

The mechanisms associated with PKM2 translocation are diverse, the most important of which is Erk1/2-dependent phosphorylation of PKM2 at serine 37 (Ser37) and FGFR1-dependent phosphorylation of PKM2 at tyrosine 105 (Tyr105) (*Li et al., 2016*; *Zhou et al., 2018*). Consequently, we demonstrated that TIPE increased the levels of PKM2 Ser37 but not PKM2 Tyr105. Furthermore, we conjectured that this transformation caused by Ser37 phosphorylation affects the interaction between TIPE and PKM2. The results showed that mutation of PKM2 at S37A blocked its interaction with TIPE but not with Y105D. Because phosphorylation of PKM2 at serine 37 is dependent on ERK1/2 (*Yang et al., 2012*) and TIPE promotes AML chemoresistance by activating the ERK signaling pathway (*Pang et al., 2020*), we used an ERK inhibitor to further demonstrate the activation of HIF-1α induced by TIPE. Not surprisingly, we found that TIPE facilitates the Warberg effect predominantly by driving PKM2 Ser37 phosphorylation. Furthermore, we demonstrated that the augmentation of TIPE on HRE promoter activation or HIF-1α targeted genes such as *LDHA* and *SLC2A1* was blocked by the S37A mutation, as previously reported. These results demonstrated that TIPE promotes HIF-1α transcriptional activation via ERK-dependent PKM2 Ser37 phosphorylation.

Increasing evidence has shown that HIF-1α promotes tumor progression, metastasis, and therapeutic drug resistance by maintaining cancer stemness (*Cui et al., 2017*; *Niu et al., 2021*). Here, we observed that TIPE significantly enhanced tumorigenesis, glycolysis, and CSC properties under normoxia in melanoma, whereas suppression of PKM2 dimerization rescued these TIPE-induced effects. Interestingly, we revealed that decreasing of PKM2 dimerization can rescue TIPE-induced cell proliferation, glycolysis, and CSC properties. However, this phenomenon was diminished by activation of HIF-1α, demonstrating that TIPE promotes melanoma cells proliferation, glycolysis, and CSC properties via dimeric PKM2-dependent HIF-1α activation. Furthermore, using a tissue ship, we showed that the expression of TIPE in melanoma tissues was dramatically increased compared to that in benign skin tumor tissues. To determine if there was a correlation between TIPE and PKM2 Ser37 levels, the results also demonstrated that TIPE expression was positively correlated with PKM2 Ser37 levels. Consistently, the expression of TIPE has a positive relationship with PKM2 Ser37, LDH, and the CSCs marker CD44 in TIPE-overexpressing cell-derived xenografts. TCGA dataset revealed that TIPE has a positive relationship with CSCs markers, including *BMI1*, *NANOG*, *NOTCH1*, and *POU5F1* in melanoma. These results further demonstrate that TIPE maintains melanoma cell stem phenotypes in a PKM2 dimerization manner.

However, we measured metabolites such as glucose, pyruvate, and lactate at steady state, the methods such as isotope tracing experiment might be more suitable for study glucose catabolism into pyruvate, as well as tracing into lactate or into the tricarboxylic acid (TCA) cycle following changes in

TIPE expression. Moreover, the relationship between TIPE levels and melanoma patient outcome is not presented in this article. One reason is that the tissue microarray lack of the survival data. Interestingly, the TCGA dataset showed a contradictory result that the higher TIPE expression has a favorable prognosis for melanoma. The detailed mechanisms will be discussed in our following article, and we hope that it might as a continuous research topic for TIPE in melanoma. In summary, we identified that TIPE serves as an oncogenic candidate for melanoma and determined that TIPE promotes the Warburg effect and maintains the stemness of melanoma. TIPE binds to PKM2, inducing its dimeric transformation by enhancing ERK-dependent PKM2 Ser37 phosphorylation, and thus activating the transcription of HIF-1α, providing tumorigenicity and cancer stem-like phenomena for melanoma progression. The newly identified TIPE/PKM2/HIF-1α axis is an ideal therapeutic target for melanoma and opens an avenue for the development of a novel strategy for precision therapy.

## Materials and methods

### Cell culture and plasmids construction

Human A375, A875, G361, MM96L, and HEK-293T cell lines were purchased from the Shanghai Institute of Cell Biology (Chinese Academy of Sciences, Shanghai, China) and were maintained in Dulbecco's modified Eagle's medium (DMEM, Gibco) supplemented with 10% fetal bovine serum (Gibco) at 37°C under 5% $CO_2$. The source of the cell lines was confirmed through short tandem repeat (STR) profiling and verified to be free of mycoplasma contamination. The TIPE overexpression lentiviral vector (TIPE), TIPE expression knockdown vector (TIPE-sh) and their control vectors were constructed by GeneChem company (Shanghai, China). HIF-1α, PKM2-Flag, TIPE-HA, and their truncated vectors were constructed in our laboratory. The mutant vectors such as PKM2 S37A were synthesized by Tsingke Biology Co, Ltd (Beijing, China). All plasmids were transfected into the cells according to the manufacturer's protocol.

### Cell proliferation, cell cycle, and colony formation assays

Cell proliferation was measured using the Cell Counting Kit-8 method (Dojindo, Shanghai, China). For the cell cycle assay, the cells were stained with propidium iodide (Sigma, USA) and were measured with a flow cytometer (FACS Aria 2, BD, USA). For the colony formation assay, cells were seeded, fixed, and stained as previously described (*Yang et al., 2022*).

### Transwell assay

Tranwell assay was performed as we previous described (*Yang et al., 2022*). Photographs of at least three randomly selected fields were captured and were counted.

### Bioinformatics, transcriptomics, and metabolomics analyses

The correlations between TIPE and HIF1A, hypoxia score, CSCs markers including *BMI1*, *NANOG*, *NOTCH1*, and *POU5F1* were estimated by using the ASSISTANT for Clinical Bioinformation website (https://www.aclbi.com/static/index.html#/). Transcriptomics and metabolomics were performed and analyzed by GeneChem company (Shanghai, China). Overexpression of TIPE into G361 cells and their control group were randomly divided into three groups to perform transcriptomic analyses. In details, RNA was extracted from cells by standard extraction methods, and then the RNA samples were strictly controlled by using Nanophotometer spectrophotometer to detect RNA purity, and Agilent 2100 Bioanalyzer to detect RNA integrity. Purified mRNA by Oligo (dT) magnetic beads then randomly interrupted by divalent cations in NEB Fragmentation Buffer, and the library is built according to the NEB common library construction method. The library was qualified and processed for Illumina sequencing. Finally, the results were subjected to gene differential expression analysis. Interfering TIPE in A375 cells and its control group were randomly divided into six groups to perform metabolomic analyses. In details, samples were extracted using the methanol/acetonitrile/aqueous solution method, followed by chromatography-mass spectrometry analysis. Finally, the data were analyzed, including univariate statistical analysis, multidimensional statistical analysis, differential metabolite screening, differential metabolite correlation analysis, KEGG pathway analysis, etc.

### ATP content, ATPase activity, pyruvate kinase activity, and lactic acid content measurement

The concentration of ATP was determined 72 hr after overexpression or interfering of TIPE by using an ATP assay kit (Beyotime, S0026) according to the manufacturer's instructions. ATPase activity was

measured by using an ATPase Assay Kit (Sigma-Aldrich, MAK113). Pyruvate kinase activity and lactate production after overexpression or interfering of TIPE were measured using a Pyruvate Kinase Activity Assay Kit (Nanjing Jiancheng, A076-1-1) and Lactic Acid Assay Kit (Nanjing Jiancheng, A019-2-1) according to the manufacturer's instruction, respectively.

## Extracellular acidification rate

The ECAR of melanoma cells was determined using the Seahorse XF extracellular flux analyzer (Agilent, Santa Clara, CA, USA). The cells were plated, washed before analysis. The cells were then treated at specific time points: glucose (10 mM), followed by oligomycin (1 μM) and 2-Deoxy-D-glucose (2-DG) (50 mM). Finally, the ECAR was measured by the Seahorse XF software.

## Chromatin immunoprecipitation quantitative PCR

Melanoma cells were cross-linked with 1% formaldehyde, lysed with sodium dodecyl sulfate buffer and sonicated. Sheared DNA was immunoprecipitated with PKM2 antibody and negative control anti-IgG, and quantified using SYBR Green Realtime PCR analysis (ABI, USA). Primer sequences are listed in *Supplementary file 1e*. Fold enrichment was calculated based on Ct as $2^{-\Delta(\Delta Ct)}$, where $\Delta Ct = Ct_{IP} - Ct_{Input}$ and $\Delta(\Delta Ct) = \Delta Ct_{antibody} - \Delta Ct_{IgG}$.

## Sphere formation assay

Melanoma cells transfected with TIPE or TIPE-sh were plated onto 24-well low attachment surface polystyrene plates (Corning, USA). Cells were grown in DMEM/F12 medium (Invitrogen, USA) for approximately 10 days supplemented with 20 ng/ml epidermal growth factor (EGF) (Bioworld, BK0026), 2% B27 (Invitrogen, 17504044), 20 ng/ml basic fibroblast growth factor (FGF) (PeproThech, #100-18B), and 1% N2 (Invitrogen, 17502048).

## Luciferase reporter assay

Human HEK-293T cells were seeded onto 48-well plates and transiently transfected with pHRE-Firefly luciferase reporter, TIPE-HA and PKM2-Flag or sh-RNA vector targeting PKM2 plasmids together with internal control reporter pTK (Thymidine kinase)-Renilla luciferase by Lipofectamine 2000 reagent (Thermo, USA). FLuc and RLuc activities were determined using the Dual-Luciferase Assay System (Promega, USA).

## qPCR

mRNA was isolated and cDNAs were synthesized by using a Prime Script RT Reagent Kit (TaKaRa, China) according to the manufacturer's instruction. qPCR was performed using Real SYBR Mixture (CoWin Bioscience, China) on an ABI PRISM 7500 instrument (Thermo Fisher, USA). The sequences of the primers used in this study are listed in *Supplementary file 1e*.

## Duolink proximity ligation assay

Duolink In Situ PLA kit (Sigma-Aldrich, DUO92101) was performed to demonstrate the endogenous interactions between TIPE and PKM2, and to interpret how TIPE impact the interactions between HIF-1α and PKM2 according to we described previously (*Yang et al., 2022*; *Zhao et al., 2019*). Quantification analysis was performed using the ImageJ software.

## Nuclear and cytosolic fractionation

Nuclear and cytosolic fractions were prepared using NE-PER Extraction Reagents (Thermo, 78833) according to the manufacturer's recommendation. Finally, both cytosolic and nuclear subfractions were stored in –80°C prior to western blot.

## Western blot

The prepared lysates were separated by 10% sodium dodecyl sulfate–polyacrylamide gel electrophoresis and transferred onto polyvinylidene fluoride (PVDF) membranes (Bio-Rad, USA). The membranes were then blocked and incubated overnight at 4°C with the primary antibodies (as listed in *Supplementary file 1f*). After incubated with secondary antibodHead2ies, the protein bands were visualized using a Tanon 5200 Imager (Shanghai, China).

## Immunohistochemical and immunofluorescence

Melanoma tissue chip was purchased from Zhongke Guanghua (Xi'an, China) Intelligent Biotechnology Co, Ltd (#K063Me01), and the clinicopathological characteristics of melanoma specimens are summarized in *Supplementary file 1g*. IHC staining was performed as previously reported. The samples were finally analyzed by using semiquantitative scoring criteria. The staining index values (0–12) were obtained and calculated (*Yang et al., 2022*). For immunofluorescence, cells were seeded onto 24-well plates at a density of 5000 cells per well. Then, the cells were fixed using 4% formaldehyde solution and permeabilized with 0.5% Triton X-100 solution. Finally, cells were incubated overnight with antibodies prior to blocked with 1% bovine serum albumin in phosphate-buffered saline containing 0.2% Tween-20. The fluorescence images of the cells were acquired with a fluorescence microscope equipped with appropriate filter combinations.

## Co-IP and Co-IP/MS

For Co-IP, the cells were seeded, harvested, and were then washed twice with phosphate-buffered saline and digested and lysed in lysis buffer. The lysed supernatant was incubated with Tag-conjugated beads to bait the potential interacted proteins according to the manufacturer's instruction. For Co-IP/MS, the beads consequently were washed, eluted, resolved, and then stained with Coomassie brilliant blue. The bands were excised and digested with chymotrypsin and subjected to LC–MS/MS sequencing and data analysis by OBiO Technology Corp, Ltd (Shanghai, China).

## Flow cytometry analysis

Cell apoptosis was measured by using a Cell Apoptosis Analysis Kit (Engreen Biosystem Co, Ltd, Beijing, China). The cells were then incubated in serum-free medium for starvation prior to analysis and fixed using 70% ethanol for 12 hr. Consequently, cells were washed, incubated with propidium iodide, and analyzed by an FACS scanner (Aria II, BD, USA). For CD44 analyzing, the cells were seeded, harvested, and incubated with anti-CD44 fluorescent antibody. Isotype-matched human immunoglobulins served as controls. Finally, the cells were counted with the FACS scanner.

## Mouse xenograft tumor study

Xenograft tumor studies were conducted utilizing the 6-week-old male BALB/c nude mice purchased from the Shanghai SLAC Laboratory Animal Co, Ltd. The total number of mice was randomly divided into respective groups and maintained in a standard pathogen free environment. All the animal experiments were approved by the Animal Ethics Committee of Zibo Central Hospital, Binzhou Medical University (Ethical Approval Number: 2021-037). Tumor growth was monitored regularly and the length, width, and height measurements taken every 7 days for five times. In the rescue experiment (*Figure 5E*), the tumor volumes were measurements taken every 5 days for seven times. For the rescue test, 1 week after injection, mice bearing TIPE-vector were treated with TEPP-46. TEPP-46 (50 mg/kg) or vehicle control (0.9% NaCl) was administrated via intraperitoneal injection. At the end of experiment, mice were euthanized and tumors were collected. Tumors were fixed with 4% paraformaldehyde for further IHC analysis.

## Statistical analysis

GraphPad Prism (GraphPad Software, San Diego, CA) was used for statistical analyses. Two-tailed Student's *t*-tests were used to compare two groups. Linear regression analysis was explored to analyze the correlation of TIPE with other indicators. All the data are shown as the standard error of mean from at least three independent experiments. A p value of <0.05 was considered statistically significant for all tests.

## Acknowledgements

This work was supported by the National Natural Science Foundation of China (Nos. 81972002, 12304241, and 81971491), Taishan Young Scholar Foundation of Shandong Province (tsqnz20231257), Natural Science Foundation of Shandong Province (ZR2024MH175, ZR2023QC168, ZR2023MC136, ZR2021MC083, and ZR2021MC165), Medicine and Health Science Technology Development Projects

of Shandong Province (202102050803 and 2018WS534), and Xinjiang Uygur Autonomous Region Training Program of Innovation and Entrepreneurship for College Students (S202310760060).

## Additional information

### Funding

| Funder | Grant reference number | Author |
| --- | --- | --- |
| National Natural Science Foundation of China | 81972002 | Peiqing Zhao |
| National Natural Science Foundation of China | 12304241 | Maojin Tian |
| National Natural Science Foundation of China | 81971491 | Yunwei Lou |
| Taishan Young Scholar Foundation of Shandong Province | tsqnz20231257 | Maojin Tian |
| Natural Science Foundation of Shandong Province | ZR2024MH175 | Peiqing Zhao |
| Natural Science Foundation of Shandong Province | ZR2023QC168 | Maojin Tian |
| Natural Science Foundation of Shandong Province | ZR2023MC136 | Peiqing Zhao |
| Natural Science Foundation of Shandong Province | ZR2021MC083 | Peiqing Zhao |
| Natural Science Foundation of Shandong Province | ZR2021MC165 | Wei Hu |
| Medicine and Health Science Technology Development Projects of Shandong Province | 202102050803 | Peiqing Zhao |
| Medicine and Health Science Technology Development Projects of Shandong Province | 2018WS534 | Peiqing Zhao |
| Xinjiang Uygur Autonomous Region Training Program of Innovation and Entrepreneurship for College Students | S202310760060 | Ziqian Zhao |

The funders had no role in study design, data collection, and interpretation, or the decision to submit the work for publication.

### Author contributions

Maojin Tian, Data curation, Investigation; Le Yang, Ziqian Zhao, Wei Hu, Investigation; Jigang Li, Validation; Lianqing Wang, Data curation; Qingqing Yin, Software; Yunwei Lou, Visualization; Jianxin Du, Methodology, Writing – original draft; Peiqing Zhao, Conceptualization, Supervision

### Author ORCIDs

Maojin Tian ⓘ https://orcid.org/0009-0004-9302-0162
Peiqing Zhao ⓘ https://orcid.org/0000-0002-9311-1927

## Ethics

This study has been approved by the Medical Ethics Committee, the Animal Care and Use Committee of Zibo Central Hospital, Binzhou Medical University (Ethical Approval Number: 2021-037). The human tissue microarray was purchased from Bioaitech Company (Xi'an, China).

Reviewer #1 (Public review): https://doi.org/10.7554/eLife.92741.4.sa1
Reviewer #2 (Public review): https://doi.org/10.7554/eLife.92741.4.sa2
Author response https://doi.org/10.7554/eLife.92741.4.sa3

---

# Additional files

## Supplementary files

Supplementary file 1. Supplementary tables. (**a**) The top candidate interacting proteins of TIPE identified by mass spectrometry. (**b**) Genes upregulated in response to low-oxygen levels (hypoxia) in melanoma. (**c**) Limiting dilution data. (**d**) Confidence intervals for 1/(stem cell frequency). (**e**) Primer or siRNA sequences. (**f**) Experimental materials. (**g**) The clinicopathological characteristics of 48 melanoma specimens.

MDAR checklist

## Data availability

All the data needed to evaluate the conclusions of the study are presented in this paper. The generated sequencing data (transcriptomics and metabolomics data) have been deposited in Dryad (https://doi.org/10.5061/dryad.ghx3ffc05).

The following dataset was generated:

| Author(s) | Year | Dataset title | Dataset URL | Database and Identifier |
|---|---|---|---|---|
| Zhao P, Tian M DJ | 2024 | Data from: TIPE drives a cancer stem-like phenotype by promoting glycolysis via PKM2/HIF-1α axis in melanoma | https://doi.org/10.5061/dryad.ghx3ffc05 | Dryad Digital Repository, 10.5061/dryad.ghx3ffc05 |

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
