## [Editor Report · eLife Assessment]

This **important** study investigates the molecular mechanisms underpinning how the tumor necrosis factor alpha-induced protein (TIPE) regulates aerobic glycolysis to promote tumor growth in melanoma. **Convincing** data using multiple independent approaches provides new insights into the molecular mechanisms underpinning aerobic glycolysis in melanoma cells. The work will be of interest to biomedical researchers working in cancer and metabolism.

---

## [Referee Report · Reviewer #1 (Public review)]

Summary:

Tian et al. describes how TIPE regulates melanoma progression, stemness, and glycolysis. The authors link high TIPE expression to increased melanoma cell proliferation and tumor growth. TIPE causes dimerization of PKM2, as well as translocation of PKM2 to the nucleus, thereby activating HIF-1alpha. TIPE promotes the phosphorylation of S37 on PKM2 in an ERK-dependent manner. TIPE is shown to increase stem-like phenotype markers. The expression of TIPE is positively correlated with the levels of PKM2 Ser37 phosphorylation in murine and clinical tissue samples. Taken together, the authors demonstrate how TIPE impacts melanoma progression, stemness, and glycolysis through dimeric PKM2 and HIF-1alpha crosstalk.

The authors manipulated TIPE expression using both shRNA and overexpression approaches throughout the manuscript. Using these models, they provide strong evidence of the involvement of TIPE in mediating PKM2 Ser37 phosphorylation and dimerization. The authors also used mutants of PKM2 at S37A to block its interaction with TIPE and HIF-1alpha. In addition, an ERK inhibitor (U0126) was used to block the phosphorylation of Ser37 on PKM2. The authors show how dimerization of PKM2 by TIPE causes nuclear import of PKM2 and activation of HIF-1alpha and target genes. Pyridoxine was used to induce PKM2 dimer formation, while TEPP-46 was used to suppress PKM2 dimer formation. TIPE maintains stem cell phenotypes by increasing expression of stem-like markers. Furthermore, the relationship between TIPE and Ser37 PKM2 was demonstrated in murine and clinical tissue samples.

The evaluation of how TIPE causes metabolic reprogramming can be further assessed using isotope tracing experiments.

---

## [Referee Report · Reviewer #2 (Public review)]

In this article, Tian et al present a convincing analysis of the molecular mechanisms underpinning TIPE-mediated regulation of glycolysis and tumor growth in melanoma. The authors begin by confirming TIPE expression in melanoma cell lines and identify "high" and "low" expressing models for functional analysis. They show that TIPE depletion slows tumour growth in vivo, and using both knockdown and over expression approaches, show that this is associated with changes in glycolysis in vitro. Compelling data using multiple independent approaches is presented to support an interaction between TIPE and the glycolysis regulator PKM2, and over-expression of TIPE promoted nuclear translocation of PKM2 dimers. Mechanistically, the authors also demonstrate that PKM2 is required for TIPE-mediated activation of HIF1a transcriptional activity, as assessed using an HRE-promoter reporter assay, and that TIPE-mediated PKM2 dimerization is p-ERK dependent. Finally, the dependence of TIPE activity on PKM2 dimerization was demonstrated on tumor growth in vivo and in regulation of glycolysis in vitro, and ectopic expression of HIF1a could rescue inhibition of PKM2 dimerization in TIPE overexpressing cells and reduced induction of general cancer stem cell markers, showing a clear role for HIF1a in this pathway.

The detailed mechanistic analysis of TIPE mediated regulation of PKM2 to control aerobic glycolysis and tumor growth is a major strength of the study and provides new insights into the molecular mechanisms that underpin the Warburg effect in melanoma cells. The main conclusions of this paper are well supported by data, however further investigation of a potential oncogenic effect of TIPE in melanoma patients is warranted to support the tumor promoting role of TIPE identified in the experimental models. Analysis of patient samples showed a significant increase in TIPE protein levels in primary melanoma compared to benign skin tumours, and a further increase upon metastatic progression. Moreover, TIPE levels correlate with proliferation (Ki67) and hypoxia gene sets in the TCGA melanoma patient dataset. However, intriguingly, high TIPE expression associates with better survival outcomes in the TCGA melanoma patient cohort, therefore further investigation of how TIPE-mediated regulation of glycolysis contributes to melanoma progression is warranted to confirm the authors claims of a potential oncogenic function. Regardless, the new insights into the molecular mechanisms underpinning TIPE-mediated aerobic glycolysis in melanoma are convincing and will likely generate interest in the cancer metabolism field.

---

## [Author Response]

The following is the authors’ response to the previous reviews.

**Reviewer #1 (Public Review):**
Summary:Tian et al. describe how TIPE regulates melanoma progression, stemness, and glycolysis. The authors link high TIPE expression to increased melanoma cell proliferation and tumor growth. TIPE causes dimerization of PKM2, as well as translocation of PKM2 to the nucleus, thereby activating HIF-1alpha. TIPE promotes the phosphorylation of S37 on PKM2 in an ERK-dependent manner. TIPE is shown to increase stem-like phenotype markers. The expression of TIPE is positively correlated with the levels of PKM2 Ser37 phosphorylation in murine and clinical tissue samples. Taken together, the authors demonstrate how TIPE impacts melanoma progression, stemness, and glycolysis through dimeric PKM2 and HIF-1alpha crosstalk.Strengths:The authors manipulated TIPE expression using both shRNA and overexpression approaches throughout the manuscript. Using these models, they provide strong evidence of the involvement of TIPE in mediating PKM2 Ser37 phosphorylation and dimerization. The authors also used mutants of PKM2 at S37A to block its interaction with TIPE and HIF-1alpha. In addition, an ERK inhibitor (U0126) was used to block the phosphorylation of Ser37 on PKM2. The authors show how dimerization of PKM2 by TIPE causes nuclear import of PKM2 and activation of HIF-1alpha and target genes. Pyridoxine was used to induce PKM2 dimer formation, while TEPP-46 was used to suppress PKM2 dimer formation. TIPE maintains stem cell phenotypes by increasing the expression of stem-like markers. Furthermore, the relationship between TIPE and Ser37 PKM2 was demonstrated in murine and clinical tissue samples.Weaknesses:The evaluation of how TIPE causes metabolic reprogramming can be better assessed using isotope tracing experiments and improved bioenergetic analysis.

Thank you immensely for your invaluable suggestions. Regrettably, we encountered a significant obstacle in completing the isotope tracing experiments due to an unfortunate shortage of necessary instruments. Furthermore, despite our efforts to consult with several companies, we were unable to secure their assistance, which unfortunately hindered the completion of these experiments. We deeply apologize for this imperfection in our experimental design and have thoroughly discussed this limitation in our manuscript.

Additionally, we acknowledge our oversight in the previous versions of our manuscripts, where only three metabolites were presented. To rectify this and provide a more comprehensive understanding of the metabolic reprogramming induced by TIPE, we have conducted routine untargeted metabolomics analysis. We are pleased to announce that we have incorporated the detailed results of this analysis into our work as a new supplementary figure, designated as Figure S3. This figure specifically highlights the notable decrease in the glycolysis pathway, particularly in pyruvate and lactic acid levels, following TIPE interference.

**Reviewer #2 (Public Review):**
In this article, Tian et al present a convincing analysis of the molecular mechanisms underpinning TIPE-mediated regulation of glycolysis and tumor growth in melanoma. The authors begin by confirming TIPE expression in melanoma cell lines and identify "high" and "low" expressing models for functional analysis. They show that TIPE depletion slows tumour growth in vivo, and using both knockdown and over-expression approaches, show that this is associated with changes in glycolysis in vitro. Compelling data using multiple independent approaches is presented to support an interaction between TIPE and the glycolysis regulator PKM2, and the over-expression of TIPE-promoted nuclear translocation of PKM2 dimers. Mechanistically, the authors also demonstrate that PKM2 is required for TIPE-mediated activation of HIF1a transcriptional activity, as assessed using an HRE-promoter reporter assay, and that TIPE-mediated PKM2 dimerization is p-ERK dependent. Finally, the dependence of TIPE activity on PKM2 dimerization was demonstrated on tumor growth in vivo and in the regulation of glycolysis in vitro, and ectopic expression of HIF1a could rescue the inhibition of PKM2 dimerization in TIPE overexpressing cells and reduced induction of general cancer stem cell markers, showing a clear role for HIF1a in this pathway. The main conclusions of this paper are well supported by data, but some aspects of the experiments need clarification and some data panels are difficult to read and interpret as currently presented.The detailed mechanistic analysis of TIPE-mediated regulation of PKM2 to control aerobic glycolysis and tumor growth is a major strength of the study and provides new insights into the molecular mechanisms that underpin the Warburg effect in cancer cells. However, despite these strengths, some weaknesses were noted, which if addressed will further strengthen the study.(1) The analysis of patient samples should be expanded to more directly measure the relationship between TIPE levels and melanoma patient outcome and progression (primary vs metastasis), to build on the association between TIPE levels and proliferation (Ki67) and hypoxia gene sets that are currently shown.

Thanks for your suggestions. We have expanded the analysis to include the relationship between TIPE levels and melanoma progression, specifically distinguishing between non-lymph node metastasis and lymph node metastasis. In addition, we added the association between TIPE and Ki67 or LDH levels as your advised, as shown in Figure 7.

However, the relationship between TIPE levels and melanoma patient outcome is not presented in this article. One reason is that the tissue microarray lack of the survival data. Interestingly, the TCGA dataset showed that the higher TIPE expression has a favorable prognosis for melanoma. We are also very curious about this. Our following study indicated that TIPE might serve as a positive regulator of PD-L1. Therefore, the higher expression of TIPE presents more sensitive tendency to immunotherapy, resulting in a favorable prognosis in melanoma. The detailed mechanisms will be discussed in our following article, and we hope that it might as a continuous research topic for TIPE in melanoma.

We just only disclose a little information that TIPE shares similar survival and immune signature to PD-L1 and PD-1 in melanoma as following:

(2) The duration of the in vivo experiments was not clearly defined in the figures, however, it was clear from the tumor volume measurements that they ended well before standard ethical endpoints in some of the experiments. A rationale for this should be provided because longer-duration experiments might significantly change the interpretation of the data. For example, does TIPE depletion transiently reduce or lead to sustained reductions in tumor growth?

Thanks for your suggestions. Actually, we have performed a pre-experiment before the formal experiments, and all the time points were referred to this. Furthermore, we have added the detailed time points into the figure legends as you suggested.

(3) The analysis of general cancer stem cell markers is solid and interesting, however inclusion of neural crest stem cell markers that are more relevant to melanoma biology would greatly strengthen this aspect of the study.

Thanks for your advices. We have selected two neural crest stem cell markers including Nestin and Sox10 to test their expression after overexpression of TIPE in G361 cells or interference of TIPE in A375 cells.

(4) The authors should take care that all data panels are clearly readable in the figures to facilitate appropriate interpretation by the reader.

Thanks for your suggestions. We have amended the data panels according to you advises to ensure it is clear and professionally presented.

**Reviewer #1 (Recommendations for the authors):**
It would be suggested to improve the image quality of certain panels (please refer to Fig.1A and Fig.S3B-D).

Thank you for your expert advice. We have optimized the quality of certain panels according to your suggestions.

**Reviewer #2 (Recommendations for the authors):**
Major comments:- TCGA survival/patient outcome data relative to TIPE levels should be provided in the supplementary figures, together with TIPE correlation with PKM2.- Suggest revising how this point is described in the discussion.

We have added the results of TIPE expression and prognosis of melanoma patients from the TCGA database as required by the expert, and discussed it appropriately in the article. In addition, the correlation between TIPE and PKM2 expression has already been described in Supplementary Figure 6.